# On-the-fly Adaptation of Patrolling Strategies in Changing Environments

**Tomáš Brázdil**    **David Klaška**    **Antonín Kučera**    **Vít Musil**    **Petr Novotný**    **Vojtěch Řehák**

Masaryk University, Faculty of Informatics, Brno, Czechia

## Abstract

We consider the problem of efficient patrolling strategy adaptation in a changing environment where the topology of Defender's moves and the importance of guarded targets change unpredictably. The Defender must instantly switch to a new strategy optimized for the new environment, not disrupting the ongoing patrolling task, and the new strategy must be computed promptly under all circumstances. Since strategy switching may cause unintended security risks compromising the achieved protection, our solution includes mechanisms for detecting and mitigating this problem. The efficiency of our framework is evaluated experimentally.

## 1 INTRODUCTION

In *patrolling games*, a *Defender* moves among vulnerable targets and strives to detect a possible ongoing attack. The targets are modeled as vertices in a directed graph, where the edges correspond to admissible moves of the Defender.

An attack at a target $\tau$ takes $d(\tau)$ time units to complete successfully. If an initiated attack is *not* discovered in the next $d(\tau)$ time units, the Defender loses a utility determined by the cost of $\tau$. The *protection value* of a Defender's strategy $\sigma$ is the expected Defender's utility guaranteed by $\sigma$ against an arbitrary Attacker's strategy.

*Adversarial* patrolling assumes a powerful Attacker who can observe Defender's moves, knows the Defender's strategy and uses this information when planning an attack. The Defender's moving strategy is typically *randomized* [Klaška et al., 2021] to prevent the Attacker from fully anticipating future moves. The adversarial setting is particularly apt when the Attacker's abilities are *unknown* and certain protection degree is required even in the worst case.

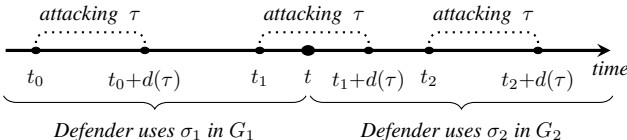

Figure 1: The coverage of an attack initiated at time $t_1$ short before the strategy switch can be very low due to the "incompatibility" of strategies $\sigma_1$ and $\sigma_2$.

Existing works focus on computing a Defender's strategy (moving plan) maximizing the protection value in a *fixed* patrolling graph. This is challenging on its own because even special variants of the problem are PSPACE-hard [Ho and Ouaknine, 2015]. However, having the underlying graph fixed is a significant limitation since the environment *does* change in real-life use cases over time and the Defender is required to adapt its strategy on-the-fly. For instance, admissible moves of a police patrol are influenced by car accidents or traffic intensity, patrolling drones are affected by weather etc. The target costs also naturally evolve; for example, the cost of a storage place decreases when emptied, etc.

When the patrolling graph $G_1$ changes into $G_2$, the current Defender's strategy $\sigma_1$ must be promptly replaced with another strategy $\sigma_2$ optimized for the new graph. In principle, $\sigma_2$ can be computed by one of the existing strategy synthesis algorithms for *fixed* patrolling graphs with $G_2$ on input. However, we show that this approach has a major *conceptual flaw*. Namely, ignoring the functionality of $\sigma_1$ when constructing $\sigma_2$ may lead to creating unnecessary *security holes* caused by the "incompatibility" of $\sigma_1$ and $\sigma_2$. Furthermore, existing algorithms for fixed patrolling graphs are *not* sufficiently efficient to be run under real-time constraints.

To understand the origin and impact of security holes, consider the scenario of Fig. 1. Here a patrolling graph $G_1$ changes into $G_2$ at time $t$, and a Defender's strategy $\sigma_1$ is replaced with $\sigma_2$. Since $\sigma_1$ and $\sigma_2$ are optimized for $G_1$ and $G_2$ respectively, they plan visits to all targets (including $\tau$)

*Accepted for the 38th Conference on Uncertainty in Artificial Intelligence* (UAI 2022).

so that the expected damage is constrained by the protection values of $\sigma_1$ and $\sigma_2$. An attack at $\tau$ initiated at time $t_0 \leq t - d(\tau)$ is fully covered by $\sigma_1$, and an attack initiated at time $t_2 \geq t$ is fully covered by $\sigma_2$. Hence, these attacks are no more dangerous than others. Now consider an attack initiated at time $t_1$ "short before" the strategy switch. If $\sigma_2$ ignores the functionality of $\sigma_1$, it may happen that $\sigma_1$ does not patrol $\tau$ in the first $t - t_1$ time units, and $\sigma_2$ omits $\tau$ in the next $d(\tau) - (t - t_1)$ time units (i.e., both strategies plan to visit $\tau$ "later"). If this happens, switching from $\sigma_1$ to $\sigma_2$ at time $t$ creates a temporary but *exceptionally dangerous* attack opportunity, i.e., a *security hole*. A simple concrete instance with quantitative analysis is given in Example 2.

Large security holes are particularly awkward when environmental changes are frequent. Regardless of their frequency, security holes compromise the protection quality and cannot be ignored when we aim at providing robust security guarantees under all circumstances. In general, the difference between $G_1$ and $G_2$ may be so large that creating security holes becomes *unavoidable* (see Example 3). This motivates the problems of algorithmic *detection*, *analysis*, and *mitigation* of security holes for a given pair of strategies $\sigma_1$ and $\sigma_2$.

The term "Defender" actually refers to the whole patrolling infrastructure, including systems for observing environmental changes, synthesizing new strategies, and deploying them to the moving agents. Hence, we assume the Defender observes environmental changes when they happen, and it has sufficient computational resources at its disposal. For the Attacker, we keep the *worst-case* approach, assuming it can observe Defender's moves, environmental changes when they happen and knows the Defender's strategies before/after the change. Furthermore, when evaluating the achieved protection, we assume the environment changes at the moment *least convenient* for the Defender. Consequently, the constructed strategies are *resistant even to sophisticated attacks when the Attacker utilizes all of this information.*

**Contribution** *We efficiently solve the problem of on-the-fly patrolling strategy adaptation in a changing adversarial environment.* Our approach overcomes the aforementioned problems and is applicable to real-world scenarios. Namely:

(1) We introduce an appropriate formal model for changing environments and strategy switching.

(2) We formalize the concept of security holes. We design an efficient algorithm for detecting and estimating security holes caused by a given strategy switch.

(3) We design an algorithm for computing a Defender's strategy $\sigma_2$ replacing the original strategy $\sigma_1$ when the underlying patrolling graph $G_1$ changes into $G_2$. This algorithm reduces the danger of creating large security holes and it is *sufficiently efficient* to be run on the fly.

(4) We show that, under certain conditions, security holes can be *mitigated* by randomized strategy switching.

(5) We confirm the efficiency of our algorithms experimentally on instances of considerable size.

As a byproduct of our effort, we obtain a strategy synthesis algorithm for fixed patrolling graphs outperforming the best existing algorithm by a margin.

Existing works on patrolling in dynamic environments are applicable to special graph topologies, non-adversarial environment, or concentrate on collaborative problems such as optimal reassigning the targets to agents (see Related Work). To the best of our knowledge, the presented results are the *first attempt* to solve the problem of *dynamic adaptation of moving strategies in adversarial changing environment with general topology*. We believe that the introduced concept of security holes is of broader interest. The underlying observations may help to handle similar issues in a larger class of dynamic planning problems with recurrent time-bounded objectives, where the new strategy is obliged to satisfy the commitments not fully accomplished by the old strategy.

Due to space constraints, some proofs are omitted. These can be found in Brázdil et al. [2022] together with outcomes of additional experiments. The code needed for reproducing the experiments is available at https://gitlab.fi.muni.cz/formela/2022-UAI-changing-env.

## 1.1 RELATED WORK

Our paper fits the *security games* line of work studying optimal allocation of limited security resources for achieving optimal target coverage [Tambe, 2011]. Practical applications of security games include the deployment of police checkpoints at the Los Angeles International Airport [Pita et al., 2008], the scheduling of federal air marshals over the U.S. domestic airline flights [Tsai et al., 2009], the arrangement of city guards in Los Angeles Metro [Fave et al., 2014], the positioning of U.S. Coast Guard patrols to secure selected locations [An et al., 2014], and also applications to wildlife protection in Uganda [Ford et al., 2014].

Most of the previous results about *adversarial patrolling games* where the Defender is mobile, the environment is actively hostile, and the game horizon is infinite concentrate on computing an optimal moving strategy for certain graph topologies. The underlying solution concept is the *Stackelberg equilibrium* [Sinha et al., 2018, Yin et al., 2010], where the Defender/Attacker play the roles of the Leader/Follower.

For general topologies, the existence of a perfect Defender's strategy discovering all attacks in time is PSPACE-complete [Ho and Ouaknine, 2015]. Consequently, computing an optimal Defender's strategy is PSPACE-hard. Moreover, computing an $\varepsilon$-optimal strategy for $\varepsilon \leq 1/2n$, where $n$ is the number of vertices, is NP-hard [Klaška et al., 2020]. Hence, no feasible strategy synthesis algorithm can *guarantee* (sub)optimality for all inputs, and finding high-quality

strategy in reasonable time is challenging. The existing methods are based on mathematical programming, reinforcement learning, or gradient descent. The first approach suffers from scalability issues caused by non-linear constraints [Basilico et al., 2012, 2009]. Reinforcement learning has so far been successful mainly for patrolling with finite horizon, such as green security games [Wang et al., 2019, Biswas et al., 2021, Xu, 2021, Karwowski et al., 2019]. Gradient descent techniques for finite-memory strategies [Kučera and Lamser, 2016, Klaška et al., 2018, 2021] are applicable to patrolling graphs of reasonable size. Strategy synthesis for restricted topologies has been studied for lines, circles [Agmon et al., 2008a,b], or fully connected environments [Brázdil et al., 2018].

Dynamically changing environments have so far been considered mainly in the context of multi-agent patrolling where the task is to dynamically reassign the targets to agents [Othmani-Guibourg et al., 2017, Seok et al., 2017, Chen et al., 2016, Hoshino and Takahashi, 2019, Das et al., 2019].

## 2 BACKGROUND

We recall the standard notions of a patrolling graph, Defender's and Attacker's strategies and their values. Since our experiments also involve comparison with state-of-the-art strategy synthesis algorithm for fixed patrolling graphs [Klaška et al., 2021], we adopt the same setup.

**Patrolling graph**    A (static) *patrolling graph* is a tuple $G = (V, T, E, time, d, \alpha)$ where

- $V$ is a finite set of *vertices* (Defender's positions);
- $T \subseteq V$ is a non-empty set of *targets*;
- $E \subseteq V \times V$ is a set of *edges* (admissible moves);
- $time \colon E \to \mathbb{N}_+$ specifies the time to travel an edge;
- $d \colon T \to \mathbb{N}_+$ assigns the time to complete an attack;
- $\alpha \colon T \to \mathbb{R}_+$ defines the costs of targets.

We write $u \to v$ instead of $(u, v) \in E$, and denote $\alpha_{\max} = \max_{\tau \in T} \alpha(\tau)$ and $d_{\max} = \max_{t \in T} d(t)$. In the sequel, let $G$ be a fixed patrolling graph.

**Defender's strategy**    In general, the Defender may choose the next vertex randomly depending on the whole history of previously visited vertices. As observed by Klaška et al. [2021], a subclass of *regular* Defender's strategies achieves the same limit protection as general strategies, and it is more convenient for algorithmic synthesis.

In the area of graph games, regular strategies are also known as *finite-memory strategies with stochastic memory update*. Intuitively, such a strategy is represented by a finite-state probabilistic automaton $\mathcal{A}$ that "reads" the sequence of vertices visited so far. When a new vertex $v$ is read, $\mathcal{A}$ changes its current state $m$ into another state $m'$ chosen randomly according to a fixed probability distribution determined by $m$ and $v$. The decision taken by the strategy then depends only on the vertex currently visited and the current state of $\mathcal{A}$. Hence, the set of states of $\mathcal{A}$, denoted by $mem$, can be seen as a finite memory where some information about the history of visited vertices is stored (we also refer to the states of $\mathcal{A}$ as *memory elements*).

Formally, let $mem$ be a finite set. The corresponding set of *augmented vertices* $\widehat{V}$ is defined as $V \times mem$, and we use $\widehat{v}$ to denote an augmented vertex of the form $(v, m)$. An *augmented edge* is a pair $\widehat{e} \equiv (\widehat{v}, \widehat{u})$ of augmented vertices where $e \equiv (v, u) \in E$. The set of all augmented edges is denoted by $\widehat{E}$.

A *regular Defender's strategy* for $G$ is a function $\sigma$ assigning to every $\widehat{v} \in \widehat{V}$ a probability distribution over $\widehat{V}$ so that $\sigma(\widehat{v})(\widehat{u}) > 0$ only if $v \to u$. Intuitively, the Defender starts in some $v \in V$ where the state of $\mathcal{A}$ is initialized to some $m \in mem$, and then it randomly selects the next vertex and the next memory element according to $\sigma$. Thus, $\sigma$ encodes both the selection of the next vertex and the choice of the next state performed by $\mathcal{A}$.

Let us fix an initial augmented vertex $\widehat{v}$. For every finite sequence $h = \widehat{v}_1, \ldots, \widehat{v}_n$, we use $\mathrm{Prob}(h)$ to denote the probability of executing $h$ under $\sigma$ when the Defender starts patrolling in $\widehat{v}$. That is, $\mathrm{Prob}(h) = 0$ if $\widehat{v} \neq \widehat{v}_1$, otherwise $\mathrm{Prob}(h) = \prod_{i=1}^{n-1} \sigma(\widehat{v}_i)(\widehat{v}_{i+1})$. Whenever we write $\mathrm{Prob}(h)$, the associated $\sigma$ and $\widehat{v}$ are clearly determined by the context.

**Attacker's strategy**    In the patrolling graph, the time is spent by traversing edges. Adversarial patrolling assumes a powerful Attacker capable of determining the next edge taken by the Defender immediately after its departure from the vertex currently visited. For the Attacker, this is an optimal moment to attack because delaying the attack gains no advantage (as we shall see, this is no longer true in a *changing* environment). Furthermore, the Attacker can attack *at most once* during a play.

An *observation* is a sequence $o = v_1, \ldots, v_n, v_n \to v_{n+1}$, where $v_1, \ldots, v_n$ is a path in $G$. Intuitively, $v_1, \ldots, v_n$ is the sequence of vertices visited by the Defender, $v_n$ is the currently visited vertex, and $v_n \to v_{n+1}$ is the edge taken next. The set of all observations is denoted by $\Omega$. An *Attacker's strategy* is a function $\pi \colon \Omega \to \{wait, attack_\tau : \tau \in T\}$. As usual, we require that if $\pi(v_1, \ldots, v_n, v_n \to u) = attack_\tau$ for some $\tau \in T$, then $\pi(v_1, \ldots, v_i, v_i \to v_{i+1}) = wait$ for all $1 \leq i < n$. Intuitively, this ensures that the Attacker can attack at most once (this assumption is standard; see, e.g., [Klaška et al., 2018, 2021] for a more detailed explanation).

**Evaluating Defender's strategy**    Let $\sigma$ be a regular Defender's strategy and $\pi$ an Attacker's strategy.

Let us fix an initial augmented vertex $\widehat{v}$ where the Defender starts patrolling. The *expected Attacker's utility* for $\sigma$, $\pi$ and $\widehat{v}$ is defined as

$$\mathrm{EAU}^{\sigma,\pi}(\widehat{v}) \;=\; \sum_{\tau,\widehat{e}} \mathbf{P}^{\sigma,\pi}(\widehat{e},\tau) \cdot \mathrm{Steal}^{\sigma}(\widehat{e},\tau)$$

where $\mathbf{P}^{\sigma,\pi}(\widehat{e},\tau)$ is the probability of initiating an attack at $\tau$ when the Defender starts moving along $\widehat{e}$, and $\mathrm{Steal}^{\sigma}(\widehat{e},\tau)$ is the expected cost "stolen" by this attack.

More precisely, let $Att(\pi,\widehat{e},\tau)$ be the set of all $(\widehat{v}_1,\ldots,\widehat{v}_{n+1})$ such that $\pi(v_1,\ldots,v_n,v_n{\to}v_{n+1}) = \tau$ and $\widehat{e} = \widehat{v}_n \to \widehat{v}_{n+1}$. We put

$$\mathbf{P}^{\sigma,\pi}(\widehat{e},\tau) = \sum_{h\in Att(\pi,\widehat{e},\tau)} \mathrm{Prob}(h).$$

Furthermore, let $\mathbf{M}^{\sigma}(\widehat{e},\tau)$ be the probability of missing (i.e., not visiting) an augmented vertex of the form $\widehat{\tau}$ in the first $d(\tau) - time(e)$ time units by a Defender's walk initiated in $\widehat{u}$, where $\widehat{u}$ is the destination of $\widehat{e}$. We define $\mathrm{Steal}^{\sigma}(\widehat{e},\tau) = \alpha(\tau) \cdot \mathbf{M}^{\sigma}(\widehat{e},\tau)$.

Intuitively, $\mathrm{EAU}^{\sigma,\pi}(\widehat{v})$ is the expected amount "stolen" by the Attacker. The Defender and Attacker aim to minimize and maximize $\mathrm{EAU}^{\sigma,\pi}(\widehat{v})$, respectively. The *Attacker's value of $\sigma$ in $\widehat{v}$* is the expected Attacker's utility achievable when the Defender commits to $\sigma$ and starts patrolling in $\widehat{v}$, i.e., $\mathrm{AVal}_G(\sigma)(\widehat{v}) = \sup_\pi \mathrm{EAU}^{\sigma,\pi}(\widehat{v})$. The Defender can choose the initial $\widehat{v}$, and hence we also define the *Attacker's value of $\sigma$* as

$$\mathrm{AVal}_G(\sigma) = \min_{\widehat{v}} \mathrm{AVal}_G(\sigma)(\widehat{v}).$$

The *Defender's value* (or simply the *value*) is defined by

$$\mathrm{DVal}_G(\sigma)(\widehat{v}) = \alpha_{\max} - \mathrm{AVal}_G(\sigma)(\widehat{v})$$
$$\mathrm{DVal}_G(\sigma) = \alpha_{\max} - \mathrm{AVal}_G(\sigma).$$

Intuitively, $\mathrm{DVal}_G(\sigma)$ corresponds to the protection guaranteed by $\sigma$ against an arbitrary Attacker's strategy. We omit the '$G$' subscript if it is clear from the context.

## 3 CHANGING ENVIRONMENT

In this section, we introduce a formal model of changing environments, formalize the concept of strategy switching, and show how to evaluate a switching strategy in a changing environment.

We consider two types of environmental changes: *topological changes* influencing the admissible Defender's moves, i.e., inserting/deleting edges or modifying edge traversal time, and *utility changes* modifying the targets costs.

Formally, a *changing environment* is a pair $G_1 \mapsto G_2$ where $G_1$ and $G_2$ are patrolling graphs with the same set of vertices

$V$, the same set of targets $T$, and the same $d$ specifying the attack times. We write $E_i$, $time_i$, and $\alpha_i$ to denote the edges, traversal times, and target costs of $G_i$ for $i \in \{1,2\}$.

Note that our definition does not allow changing the vertex set or the target set, yet these changes can be easily modeled. For instance, adding a vertex may be modeled so that the vertex is present in both $G_1$ and $G_2$ but has no incoming edges in $G_1$ ($\sigma_1$ will be extended with an arbitrary behavior at the vertex). Similarly, removing a target may be modeled by changing its cost to a negligibly small value.

For the rest of this section, we fix a changing environment $G_1 \mapsto G_2$, and a pair of regular Defender's strategies $\sigma_1$ and $\sigma_2$ for $G_1$ and $G_2$, respectively. We assume that $\sigma_1$ and $\sigma_2$ use the same set $mem$ of memory elements.

**Strategy switching**   Let $t \in \mathbb{N}$ be a *switching time*. We use $G_1 \mapsto_t G_2$ to denote the scenario where the patrolling graph $G_1$ changes into the patrolling graph $G_2$ at time $t$, and $\sigma_1 \mapsto_t \sigma_2$ to denote the Defender's strategy for $G_1 \mapsto_t G_2$ obtained by "switching" from $\sigma_1$ into $\sigma_2$ at time $t$, defined as follows.

The Defender keeps executing $\sigma_1$ in all augmented vertices visited strictly before time $t$. Let $(v,m)$ be the first augmented vertex visited by the Defender at or after time $t$ (observe that the $m$ is still determined by $\sigma_1$). From now on, the Defender should play according to $\sigma_2$. We distinguish three possibilities.

(a) There is $m' \in mem$ such that $\mathrm{DVal}_{G_2}(\sigma_2) = \mathrm{DVal}_{G_2}(\sigma_2)(v,m')$. Then, the Defender selects such an $m'$ and starts applying $\sigma_2$ from $(v,m')$.

(b) The condition of (a) does not hold, but there exist $(v',m')$ and a path from $v$ to $v'$ in $G_2$ such that $\mathrm{DVal}_{G_2}(\sigma_2) = \mathrm{DVal}_{G_2}(\sigma_2)(v',m')$. Then, the strategy $\sigma_1 \mapsto_t \sigma_2$ follows the selected path from $v$ to $v'$, and then starts applying $\sigma_2$ from $(v',m')$ for the selected $m'$.

(c) None of the conditions (a) and (b) holds. Then, it is *impossible* to perform a switch from $\sigma_1$ to $\sigma_2$ preserving the protection value of $\sigma_2$, and the strategy $\sigma_1 \mapsto_t \sigma_2$ is undefined.

In all scenarios considered in our experiments, Condition (a) holds for every $t$. Condition (b) corresponds to a situation when some vertex $v$ visited by $\sigma_1$ is no longer visited by $\sigma_2$. Condition (c) covers pathological cases when a "drastic" environmental change prevents switching $\sigma_1$ into $\sigma_2$ (e.g., all edges disappear). From now on, we assume that Condition (a) or (b) holds and the strategy $\sigma_1 \mapsto_t \sigma_2$ is defined.

**Remark 1.** *Our algorithm for constructing $\sigma_2$ (see Section 4.1) "adapts" $\sigma_1$ to the new environment $G_2$. Hence, the elements of $mem$ may represent similar information*

about the history of visited vertices in $\sigma_1$ and $\sigma_2$, and Condition (a) may hold even for $m' = m$. In this case, the information encoded by $m$ is passed on to $\sigma_2$ during the switch, decreasing the danger of creating large security holes.

**Evaluating a switching strategy** In static scenarios, it is safe to assume the Attacker initiates his attack when the Defender leaves a vertex (see the paragraph *Attacker's strategy* in the previous section). However, in $G_1 \mapsto_t G_2$, the Attacker *may* increase its expected utility by initiating an attack in the middle of a Defender's move. This is because a short delay may suffice for completing the attack *after* time $t$ when the target becomes more valuable, but postponing the attack to the moment when the Defender completes the move would already increase the probability of discovering the attack too much. A concrete example is given in Brázdil et al. [2022].

Technically, we define an *observation* in $G_1 \mapsto_t G_2$ as a pair $(o, \delta)$, where $o = v_1, \ldots, v_n, v_n \rightarrow v_{n+1}$ is defined as for static environments and $\delta \in \mathbb{N}$ is a *delay* strictly smaller than $time_i(v_n \rightarrow v_{n+1})$, where $i = 1$ if the move $v_n \rightarrow v_{n+1}$ is initiated before time $t$, and $i = 2$ otherwise.

The expected Attacker's utility $\mathrm{EAU}^{\sigma_1 \mapsto_t \sigma_2, \pi}(\widehat{v})$ is defined similarly as for static environments, i.e., as a sum

$$\sum_{\tau, \widehat{e}, \delta, t_0} \mathbf{P}^{\sigma_1 \mapsto_t \sigma_2, \pi}(\widehat{e}, \tau, \delta, t_0) \cdot \mathrm{Steal}^{\sigma_1 \mapsto_t \sigma_2}(\widehat{e}, \tau, \delta, t_0) \quad (1)$$

Here, $t_0 \in \mathbb{N}$ denotes the attack time. The symbol $\mathbf{P}^{\sigma_1 \mapsto_t \sigma_2, \pi}(\widehat{e}, \tau, \delta, t_0)$ is the probability of initiating an attack at $\tau$ at time $t_0$ when the Defender has been going along $\widehat{e}$ for $\delta$ time units (note that this also depends on the Defender's initial position $\widehat{v}$). $\mathrm{Steal}^{\sigma_1 \mapsto_t \sigma_2}(\widehat{e}, \tau, \delta, t_0)$ denotes the expected cost "stolen" by this attack. Detailed technical definitions are in Brázdil et al. [2022]. Although the delay $\delta$ further complicates our technical definitions, it describes a *real phenomenon* which must be properly reflected by a realistic formal model.

The time $t$ when $G_1$ changes into $G_2$ is unpredictable, and the strategy $\sigma_2$ must guarantee a reasonable protection on $G_2$ for all $t$'s. Hence, the *Attacker's value of $\sigma_1 \mapsto \sigma_2$ in $G_1 \mapsto G_2$* is defined as

$$\mathrm{AVal}_{G_1 \mapsto G_2}(\sigma_1 \mapsto \sigma_2) = \min_{\widehat{v}} \sup_{\pi} \sup_{t} \mathrm{EAU}^{\sigma_1 \mapsto_t \sigma_2, \pi}(\widehat{v}).$$

Note that the "$\sup_\pi$" in the above definition ensures that *all* Attacker's strategies are taken into account, including those taking advantage of observing the environmental change, Defender's moves, and analyzing the functionality of $\sigma_1, \sigma_2$.

# 4 SECURITY HOLES

Strategy switching may result in temporarily decreasing the protection of some targets. Clearly, the Defender cannot

protect $G_1 \mapsto G_2$ by $\sigma_1 \mapsto \sigma_2$ better than it is protecting $G_1$ by $\sigma_1$ and $G_2$ by $\sigma_2$. In terms of Attacker's values,

$$\mathrm{AVal}_{G_1 \mapsto G_2}(\sigma_1 \mapsto \sigma_2) \geq \mathrm{AVal}_{G_1}(\sigma_1),$$
$$\mathrm{AVal}_{G_1 \mapsto G_2}(\sigma_1 \mapsto \sigma_2) \geq \mathrm{AVal}_{G_2}(\sigma_2).$$

The first inequality is simple because the switching time $t$ can be arbitrarily large. With increasing $t$, the Attacker can perform more and more of his attacks scheduled by a given strategy $\pi$ against $\sigma_1$ in $G_1$ also in $G_1 \mapsto G_2$, achieving the expected utility arbitrarily close to the expected utility received in $G_1$. The second inequality is also immediate because the Attacker can "simulate" an arbitrary strategy $\pi$ against $\sigma_2$ also in $G_1 \mapsto G_2$ by performing his attacks after the switching time.

However, it may also happen that $\mathrm{AVal}_{G_1 \mapsto G_2}(\sigma_1 \mapsto \sigma_2)$ is *strictly larger* than the maximum of $\mathrm{AVal}_{G_1}(\sigma_1)$ and $\mathrm{AVal}_{G_2}(\sigma_2)$ due to the new attack opportunities offered in the limited time window short before the switching time caused by the "incompatibility" of $\sigma_1$ and $\sigma_2$ (see Fig. 1). Note that although the Attacker cannot enforce an environmental change at a particular time, in our adversarial setting we consider the worst possibility, i.e., we assume the change happens in the least convenient moment. Formally, the *security hole* of $\sigma_1 \mapsto \sigma_2$, denoted by $Hole_{G_1 \mapsto G_2}(\sigma_1, \sigma_2)$, is defined as

$$\mathrm{AVal}_{G_1 \mapsto G_2}(\sigma_1 \mapsto \sigma_2) - \max\{\mathrm{AVal}_{G_1}(\sigma_1), \mathrm{AVal}_{G_2}(\sigma_2)\}.$$

Intuitively, $Hole_{G_1 \mapsto G_2}(\sigma_1, \sigma_2)$ is the extra amount stolen by the Attacker due to the incompatibility of $\sigma_1$ and $\sigma_2$. Note that if $Hole_{G_1 \mapsto G_2}(\sigma_1, \sigma_2) = 0$, then the new attack opportunities caused by the switch are no more dangerous than the ones offered by $\sigma_1$ in $G_1$ and $\sigma_2$ in $G_2$.

**Example 2.** *Let $G_1$ and $G_2$ be the patrolling graphs of Fig. 2. Let $\sigma_1$ be a trivial strategy walking among $v_1, v_2, v_3$ clockwise. Since every target is revisited within the next 6 time units, all attacks are discovered in time and hence $\mathrm{AVal}_{G_1}(\sigma_1) = 0$. When the environment changes into $G_2$ by removing the edge $v_2 \rightarrow v_3$, the Defender's strategy is changed into $\sigma_2$ walking among $v_1, v_2, v_3$ anti-clockwise. Clearly, $\mathrm{AVal}_{G_2}(\sigma_2) = 0$, and hence both $\sigma_1$ and $\sigma_2$ achieve perfect protection in $G_1$ and $G_2$, resp.*

*Now consider the scenario where $v_3$ is attacked at time $\ell$ when the Defender is in the middle of the move $v_3 \rightarrow v_1$ in $G_1$. The Defender arrives in $v_1$ at time $\ell + 1$, and the environment changes from $G_1$ into $G_2$ at time $t = \ell + 2$. In $v_1$, the Defender still uses $\sigma_1$ to determine the next move, and arrives in $v_2$ at time $\ell + 3$. In $v_2$, the Defender already uses the new strategy $\sigma_2$, and therefore visits $v_3$ at time $\ell + 7$. That is, the attack at $v_3$ initiated at time $\ell$ (short before the switching time) succeeds with probability one. Consequently, $\mathrm{AVal}_{G_1 \mapsto G_2}(\sigma_1 \mapsto \sigma_2) = 100$ and hence $Hole_{G_1 \mapsto G_2}(\sigma_1, \sigma_2) = 100$.*

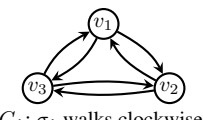
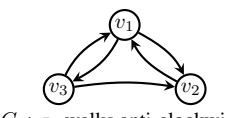

$G_1$; $\sigma_1$ walks clockwise.     $G_2$; $\sigma_2$ walks anti-clockwise.

Figure 2: Exemplifying $Hole_{G_1 \mapsto G_2}(\sigma_1, \sigma_2) = 100$. Target costs are 100, traversing every edge takes 2 time units, completing an attack takes 6 time units. If the Attacker attacks $v_3$ when the Defender is in the middle of the edge $v_3 \to v_1$ and the environment changes in 2 more time units, the attack succeeds with probability 1.

## 4.1 ESTIMATING SECURITY HOLES

Now we present an algorithm for computing an upper bound on the security hole. First, we reduce the estimation of the security hole to the computation of certain steals. Second, we make some observations allowing for considering fewer steals. Third, we present Algorithm 1, where several steals are computed at once by performing a search through the patrolling graph.

**Reduction to steals** Let $G_1, G_2$ be patrolling graphs and $\sigma_1, \sigma_2$ be Defender's strategies in $G_1$ and $G_2$, respectively. Recall that $Hole_{G_1 \mapsto G_2}(\sigma_1, \sigma_2)$ is defined as

$$\mathrm{AVal}_{G_1 \mapsto G_2}(\sigma_1 \mapsto \sigma_2) - \max\{\mathrm{AVal}_{G_1}(\sigma_1), \mathrm{AVal}_{G_2}(\sigma_2)\}.$$

Since $\mathrm{AVal}_{G_1}(\sigma_1)$ and $\mathrm{AVal}_{G_2}(\sigma_2)$ are computable by the standard strategy evaluation algorithm [see, e.g., Klaška et al., 2021], we only need to compute an *upper bound on* $\mathrm{AVal}_{G_1 \mapsto G_2}(\sigma_1 \mapsto \sigma_2)$. Recall that the expected protection achieved by $\sigma_1 \mapsto \sigma_2$ against a given attack is fully determined by the following:

- $\widehat{e}$: the Defender's location when the attack is initiated;
- $\tau$: the attacked target;
- $\delta$: the time passed since the Defender entered $\widehat{e}$;
- $t_0$: the current time (when the attack is initiated);
- $t$: the switching time.

According to the definition of security hole, it suffices to compute the maximum of all the values $\mathrm{Steal}^{\sigma_1 \mapsto_t \sigma_2}(\widehat{e}, \tau, \delta, t_0)$. Since there are *infinitely many* $t, t_0 \in \mathbb{N}$, this task is not trivial.

**Minimizing the number of steals to consider** If $t_0 < t - d(\tau)$ or $t_0 \geq t$, then the attack at $\tau$ is fully covered by $\sigma_1$ or $\sigma_2$, respectively. Hence, the only interesting case is when $1 \leq t - t_0 \leq d(\tau)$. Although there are still infinitely many $t, t_0$ satisfying this condition, the above Steal is *fully determined* just by the difference $t - t_0$, and can thus be written as $\mathrm{Steal}(\widehat{e}, \tau, \delta, \Delta t)$, where $\Delta t$ denotes the difference and ranges over *finitely many values* bounded by $d(\tau)$.

Another simple observation is that $\mathrm{Steal}(\widehat{e}, \tau, \delta, \Delta t) \leq \mathrm{Steal}(\widehat{e}, \tau, 0, \min(\Delta t + \delta, d(\tau)))$. Hence, from now on, we omit the $\delta$, implicitly assuming $\delta = 0$.

Furthermore, for all $\widehat{e}, \widehat{e}'$ such that $time_1(e) \leq time_1(e')$ and both $\widehat{e}$ and $\widehat{e}'$ lead to the *same* augmented vertex, we have that $\mathrm{Steal}(\widehat{e}, \tau, \Delta t) \leq \mathrm{Steal}(\widehat{e}', \tau, \Delta t)$. Therefore, it suffices to pick, for each augmented vertex $\widehat{v}$, one of the longest augmented edges leading to $\widehat{v}$, and disregard all other augmented edges when looking for the maximal $\mathrm{Steal}(\widehat{e}, \tau, \Delta t)$.

Finally, for given $\widehat{e}$, we say that $\Delta t$ is an *arrival time* if the Defender can reach some vertex in *precisely* $\Delta t$ time units after it starts moving along $\widehat{e}$. Note that if $\Delta t < d(\tau)$ is *not* an arrival time for $\widehat{e}$, then $\mathrm{Steal}(\widehat{e}, \tau, \Delta t) = \mathrm{Steal}(\widehat{e}, \tau, \Delta t + 1)$. Therefore, we may safely disregard all $\Delta t < d(\tau)$ that are not arrival times for $\widehat{e}$, and compute the Steal either for the least $\Delta t' > \Delta t$ which is an arrival time, or for $\Delta t' = d(\tau)$. By incorporating this condition, we obtain a set of all *eligible* $\mathrm{Steal}(\widehat{e}, \tau, \Delta t)$.

**Computing the steals** For all eligible $\widehat{e}_0$ and $\tau$, we merge the computation of $\mathrm{Steal}(\widehat{e}_0, \tau, \Delta t)$ for all eligible $\Delta t$ into one as presented in Algorithm 1: Let $\widehat{e}_0 = ((v_a, m_a), (v_b, m_b))$. If $v_b = \tau$, the answer is trivial: the Defender either surely catches the attack (if $time_1(e_0) \leq d(\tau)$) or surely fails to catch it (otherwise). Otherwise, the algorithm preforms a forward search through the patrolling graph. The search is guided by a min-heap $\mathcal{H}$ of items $(v, m, t, p)$, sorted by $t$, where each item corresponds to a certain set of paths from $(v_b, m_b)$ to $(v, m)$, all of which have the same length (total traversal time) $t$ and whose total probability is $p$. The first heap item is $(v_b, m_b, time_1(e_0), 1)$, corresponding to the Defender being at time $time_1(e_0)$ in the augmented vertex $(v_b, m_b)$ with probability 1.

Now, we explain lines 10–14. There, we compute $\mathrm{Steal}(\widehat{e}_0, \tau, \Delta t)$ for $\Delta t = \ell$. Note that the contents of $\mathcal{H}$ fully describe the possible locations of the Defender at time $\ell$: each item $h \in \mathcal{H}$ corresponds to the Defender being with probability $h.p$ on an edge leading to $(h.v, h.m)$, arriving there at time $h.t$ (if $h.t = \ell$, then the Defender is already in $(h.v, h.m)$), while failing to have caught the ongoing attack at $\tau$ yet. (Then, $1 - \sum_{h \in \mathcal{H}} h.p$ is the probability that the attack has already been caught.) Thus, it suffices to compute, for every $h \in \mathcal{H}$, the probability $p_{catch}(h)$ of visiting $\tau$ from $(h.v, h.m)$ in $G_2$ within $d(\tau) - h.t$ time units. Then, $\mathrm{Steal}(\widehat{e}, \tau, \ell) = \alpha(\tau) \cdot (\sum_{h \in \mathcal{H}} h.p \cdot (1 - p_{catch}(h)))$.

**Answering the queries for $p_{catch}(h)$** At line 12, Algorithm 1 needs to know the probability $p_{catch}(h)$. Presumably, $p_{catch}(h)$ could be computed simply by performing another similar search from $(h.v, h.m)$ in $G_2$ (omitting lines 10–14). However, this would be rather slow. Instead, we initiate a backward search from $\tau$ in $G_2$, and we make further

**input** : Patrolling graphs $G_1, G_2$, regular strategies $\sigma_1, \sigma_2, \widehat{e}_0 = ((v_a, m_a), (v_b, m_b)) \in \widehat{E}, \tau \in T$

**output** : Maximum of $\text{Steal}(\widehat{e}_0, \tau, \Delta t)$ over all $\Delta t$

```
1  V : array indexed by eligible pairs V̂
2  H : min-heap of tuples (v, m, t, p) sorted by t
3  if v_b = τ then
4  │   return time₁(e₀) ≤ d(τ) ? 0 : α(τ)
5  end
6  steal = 0
7  H.insert(v_b, m_b, time₁(e₀), 1)
8  while not H.empty do
9  │   ℓ = H.peek.t
10 │   prob = 0
11 │   foreach h ∈ H do
12 │   │   prob += h.p * (1 − Query_p_catch(h, G₂, σ₂, τ))
13 │   end
14 │   steal = max(steal, α(τ) * prob)
15 │   repeat
16 │   │   (v, m, t, p) = H.pop
17 │   │   V(v, m) += p
18 │   until H.empty or H.peek.t > ℓ
19 │   foreach (v, m) such that V(v, m) > 0 do
20 │   │   foreach ê = ((v, m), (v′, m′)) ∈ Ê do
21 │   │   │   t = ℓ + time₁(e)
22 │   │   │   if t ≤ d(τ) and v′ ≠ τ then
23 │   │   │   │   H.insert(v′, m′, t, V(v, m) * σ₁(ê))
24 │   │   end
25 │   │   V(v, m) = 0
26 │   end
27 end
28 return steal
```

**Algorithm 1:** Computes max $\text{Steal}(\widehat{e}_0, \tau, \Delta t)$ over all $\Delta t$ for a given $\widehat{e}_0$ and $\tau$

enhancements in order to answer the queries for $p_{catch}(h)$ efficiently. The details are presented in Brázdil et al. [2022].

## 4.2 PREVENTING SECURITY HOLES

Our approach to preventing large security holes is based on taking the functionality of $\sigma_1$ into account when computing the strategy $\sigma_2$ for a given $G_1 \mapsto G_2$. This is achieved by *adapting* the strategy $\sigma_1$ to $G_2$. Since $\sigma_1$ may not be directly executable in $G_2$ (for example, some edges of $G_1$ used by $\sigma_1$ may disappear in $G_2$), we first perform some adjustments to $\sigma_1$. Then, we *improve* this initial strategy in $G_2$ by an *efficient strategy improvement algorithm* described below, and thus obtain $\sigma_2$. Intuitively, since $\sigma_2$ tends to be "similar" to $\sigma_1$, the chance of producing *unnecessary* security holes decreases. This intuition is confirmed in Experiments.

The starting point for designing our strategy improvement algorithm is REGSTAR, currently the best strategy synthesis algorithm for fixed patrolling graphs presented in Klaška et al. [2021]. REGSTAR repeatedly picks a random initial strategy and tries to improve its value. The algorithm con-

sists of two subroutines: *Evaluation*, i.e., computing the value and the gradient of a given strategy, and *Optimization* using gradient descent. After hundreds of trials, the best strategy found is chosen. However, the percentage of trials converging to the best strategy found can be rather low ($\approx 2\%$) [see Klaška et al., 2021, Sec. 3.5]. For this reason, our initial attempt to construct $\sigma_2$ by applying the strategy-improvement subroutine of REGSTAR to $\sigma_1$ in the graph $G_2$ *failed*. This calls for REGSTAR re-design.

First, we replace the optimization scheme using dedicated tools for differentiable programming (PyTorch with Adam optimizer). We also add decaying Gaussian noise to the gradient allowing for different outcomes when optimizing from $\sigma_1$, improving the chance of hitting a high-valued $\sigma_2$. In contrast, the optimization loop of REGSTAR is purely deterministic. Furthermore, the REGSTAR's evaluation computes gradients in forward mode. We re-design this part by employing the reverse mode, yielding improvement by a factor of $|\widehat{E}|$.

Our modifications drastically improve REGSTAR's convergence ratio and speed (see the analysis in Section 5.1) and allow for *on-the-fly strategy adaptation*. Implementation details are documented, and the code is provided in https://gitlab.fi.muni.cz/formela/2022-UAI-changing-env.

## 4.3 MITIGATING SECURITY HOLES

In general, the structural difference between $G_1$ and $G_2$ in a changing environment $G_1 \mapsto G_2$ can make the creation of security holes *unavoidable*, as demonstrated by the following example.

**Example 3.** *Consider the setup of Example 2. There is only one $\sigma_2$ such that $\text{DVal}_{G_2}(\sigma_2) = 100$ (the "anticlockwise walk"), and hence there is no reasonable alternative to $\sigma_2$. Since $\text{DVal}_{G_1}(\sigma_1) = \text{DVal}_{G_2}(\sigma_2) = 100$, we* inevitably *obtain the largest conceivable security hole equal to* 100.

However, we show that under the conditions, the security holes can be *mitigated without harming the protection achieved by $\sigma_2$* by *randomized strategy switching*. Let us assume the following:

- For every $(v, m)$ visited by $\sigma_1$ with positive probability, there is $(v, m')$ such that $\text{DVal}_{G_2}(\sigma_2)(v, m') = \text{DVal}_{G_2}(\sigma_2)$.

- $\sigma_1$ is executable in $G_2$ and[1] $Hole_{G_1 \mapsto G_2}(\sigma_1, \sigma_1) = 0$.

Under these conditions, the Defender may perform a *randomized switch from $\sigma_1$ to $\sigma_2$*. That is, the Defender flips

---

[1]This is *not* a typo. If $G_2$ has the same topology as $G_1$ but edge traversal times change, then setting $\sigma_2 = \sigma_1$ *may* cause a security hole. The assumption $Hole_{G_1 \mapsto G_2}(\sigma_1, \sigma_1) = 0$ says that this does not happen.

a $\kappa$-biased coin when it arrives in a vertex and switches to $\sigma_2$ only with probability $\kappa$. With the remaining probability $1 - \kappa$, the Defender continues executing $\sigma_1$ and flipping the coin in the next vertex again. This goes on until the switch to $\sigma_2$ is performed. We have the following result. A proof can be found in Brázdil et al. [2022].

**Theorem 4.** *The expected number of time units needed to perform the $\kappa$-randomized switch is bounded by $max\text{-}time_2/\kappa$, where $max\text{-}time_2$ is the maximal traversal time of an edge in $G_2$. The security hole caused by the switch is bounded by*

$$\varrho + \left(1 - (1 - \kappa)^{d_{\max}}\right) \cdot \alpha_{\max}(G_2)$$

*where $\alpha_{\max}(G_2)$ is the maximal target cost in $G_2$ and $\varrho$ is defined as*

$$\max\left\{0, \, \mathrm{AVal}_{G_2}(\sigma_1) - \max\{\mathrm{AVal}_{G_1}(\sigma_1), \mathrm{AVal}_{G_2}(\sigma_2)\}\right\}.$$

*Hence, the security hole can be pushed arbitrarily close to $\varrho$ by choosing a suitably small $\kappa > 0$.*

# 5 EXPERIMENTS

## 5.1 STRATEGY IMPROVEMENT ANALYSIS

We assess our strategy synthesis algorithm in comparison with REGSTAR on the set of patrolling graphs used to evaluate REGSTAR by Klaška et al. [2021]. These graphs model office buildings, and their structure is recalled in Brázdil et al. [2022].

Here we present the outcomes for a graph modeling a 2-floor building achieved for $mem$ with $1, \ldots, 8$ elements, cf. [Klaška et al., 2021, Experiment 5.3]. Fig. 3 shows box-plot statistics of values of strategies found by 200 trials of REGSTAR (blue) and our improved (red) method. Note that our method consistently produces values concentrated around the best value found, i.e., the chance of producing a strategy with a high value from a random initial strategy is *high*. Furthermore, the value of the *best* strategy found by our method is *higher* than the one found by REGSTAR in all cases except for $|mem| = 8$ (where the difference is negligible). Similar results are obtained for *all* patrolling graphs analyzed by Klaška et al. [2021]. These datasets together with a detailed setup description are in Brázdil et al. [2022].

Next, we report runtimes of the forward (value) and backward (gradient) computations of the strategy-evaluation module. Tab. 1 summarizes the mean of 200 passes through the strategy evaluation on the same 2-floor building graph with various $mem$ sizes. Reverse-mode gradient computation improved the backward times by *three orders of magnitude* (note that the time is given in *seconds* for REGSTAR and in *miliseconds* for our method).

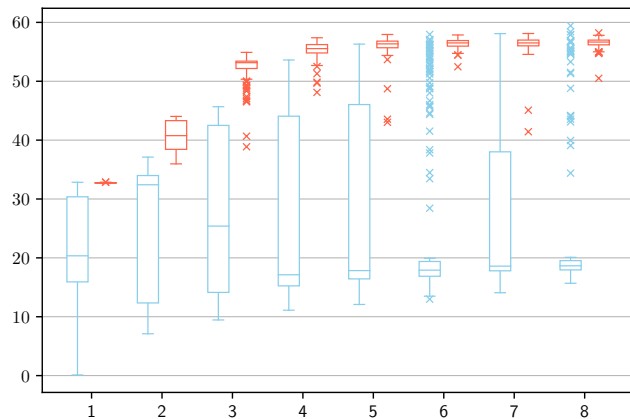

Figure 3: Values of strategies synthesized by REGSTAR (blue) and our method (red) for a 2-floor building graph where $mem$ has $1, \ldots, 8$ elements. The red values are tightly distributed close to the maxima.

| | forward [ms] | | backward | |
| $m$ | REGSTAR | Ours | REGSTAR [s] | Ours [ms] |
| --- | --- | --- | --- | --- |
| 2 | $50 \pm 3$ | $48 \pm 5$ | $0.64 \pm 0.04$ | $4 \pm 0$ |
| 4 | $217 \pm 4$ | $198 \pm 19$ | $10.2 \pm 0.7$ | $14 \pm 2$ |
| 6 | $492 \pm 4$ | $451 \pm 44$ | $57.4 \pm 3.9$ | $33 \pm 3$ |
| 8 | $913 \pm 9$ | $805 \pm 80$ | $186.3 \pm 12.0$ | $60 \pm 7$ |

Table 1: Effect of differentiation of $\sigma \mapsto \mathrm{DVal}(\sigma)$ in reverse mode (Ours) compared to forward mode (REGSTAR).

## 5.2 CHANGING ENVIRONMENT

We evaluate our algorithms for concrete changing environments. Specifically, we quantify the impact of our approach to preventing security holes and examine the effectiveness of randomized strategy switching on mitigating security holes.

We fix one patrolling graph $G_1$ consisting of 15 locations in the downtown of Vancouver. The target costs are set between 80 and 100 at random. Furthermore, we select 72 edges connecting the targets with lengths measured in taxicab distance in hundreds of meters. Attack times are fixed to 64, giving the Defender chance to discover an attack starting 6.4km far away. For $G_1$, we find and fix a strategy $\sigma_1$ with $\mathrm{DVal}_{G_1}(\sigma_1) = 42.1$.

We perform three sets of experiments, modifying $G_1$ to $G_2$ by either changing the target costs, edge lengths, or removing some edges. The experiments are parameterized by the *change size*, denoted by CS. For all types we report two CS values representing small and large change impact. More values are reported in Brázdil et al. [2022].

*Utility changes* The cost of each node is increased by its CS% with probability 1/3, decreased by CS% with probability 1/3, or left unchanged. Note that utility changes can modify $\alpha_{\max}$ and thus influence DVal. To compare, we nor-

| | CS | steps | DVal from $\sigma_1$ | DVal from rnd | Security Hole from $\sigma_1$ | Security Hole from rnd |
|---|---|---|---|---|---|---|
| utility changes | 5 | 0 | 40.9 ± 0.9 | 12.7 ± 3.2 | 0.0 ± 0.0 | 4.6 ± 2.7 |
| | | 50 | 43.4 ± 0.6 | 27.4 ± 0.8 | 2.8 ± 1.2 | 14.4 ± 1.2 |
| | | 100 | 43.6 ± 0.5 | 37.3 ± 1.2 | 3.9 ± 1.9 | 25.3 ± 2.6 |
| | | 200 | 43.8 ± 0.6 | 41.3 ± 0.6 | 5.2 ± 2.8 | 30.2 ± 4.4 |
| | | 400 | 43.8 ± 0.6 | 42.7 ± 0.5 | 6.9 ± 4.4 | 33.7 ± 3.9 |
| | 30 | 0 | 40.9 ± 0.9 | 14.7 ± 4.2 | 0.0 ± 0.0 | 6.1 ± 2.2 |
| | | 50 | 50.4 ± 3.4 | 38.2 ± 4.4 | 9.7 ± 1.9 | 19.6 ± 3.3 |
| | | 100 | 51.4 ± 3.6 | 48.7 ± 2.8 | 11.3 ± 3.1 | 30.1 ± 5.6 |
| | | 200 | 52.4 ± 3.8 | 52.8 ± 3.5 | 14.0 ± 2.3 | 39.4 ± 4.1 |
| | | 400 | 52.9 ± 3.7 | 54.0 ± 3.3 | 15.2 ± 3.2 | 40.0 ± 5.6 |
| variable edge length | 5 | 0 | 36.4 ± 2.2 | 9.4 ± 0.3 | 0.6 ± 1.0 | 2.9 ± 0.3 |
| | | 50 | 40.3 ± 0.9 | 24.3 ± 0.6 | 5.0 ± 1.9 | 13.0 ± 1.1 |
| | | 100 | 40.6 ± 0.9 | 34.7 ± 0.7 | 6.5 ± 2.6 | 26.2 ± 2.5 |
| | | 200 | 40.9 ± 0.9 | 39.3 ± 0.7 | 8.8 ± 3.1 | 31.9 ± 2.3 |
| | | 400 | 41.0 ± 0.9 | 40.4 ± 0.8 | 9.4 ± 3.1 | 33.8 ± 2.7 |
| | 30 | 0 | 16.3 ± 10.1 | 8.1 ± 1.1 | 0.1 ± 0.4 | 1.7 ± 1.2 |
| | | 50 | 33.6 ± 5.0 | 24.0 ± 2.5 | 5.4 ± 5.4 | 13.2 ± 2.8 |
| | | 100 | 36.4 ± 3.7 | 37.8 ± 2.4 | 10.2 ± 4.6 | 27.5 ± 2.3 |
| | | 200 | 38.7 ± 3.2 | 42.7 ± 2.2 | 14.2 ± 5.3 | 33.9 ± 2.6 |
| | | 400 | 40.8 ± 2.6 | 44.4 ± 2.9 | 20.9 ± 3.8 | 34.1 ± 3.5 |
| removed edges | 1 | 0 | 39.5 ± 5.3 | 9.6 ± 0.3 | 0.0 ± 0.0 | 3.1 ± 0.4 |
| | | 50 | 42.0 ± 0.2 | 24.6 ± 0.3 | 0.9 ± 1.5 | 13.3 ± 0.8 |
| | | 100 | 42.1 ± 0.1 | 35.1 ± 0.7 | 0.9 ± 1.5 | 24.9 ± 2.1 |
| | | 200 | 42.1 ± 0.1 | 39.4 ± 0.3 | 0.9 ± 1.4 | 31.9 ± 3.3 |
| | | 400 | 42.1 ± 0.1 | 40.9 ± 0.5 | 1.3 ± 1.2 | 34.5 ± 2.3 |
| | 8 | 0 | 17.0 ± 11.4 | 9.8 ± 0.9 | 0.0 ± 0.0 | 3.3 ± 0.6 |
| | | 50 | 36.1 ± 9.1 | 24.8 ± 0.3 | 8.2 ± 5.3 | 13.2 ± 1.7 |
| | | 100 | 39.7 ± 1.9 | 35.3 ± 0.9 | 10.3 ± 5.1 | 25.5 ± 2.4 |
| | | 200 | 40.4 ± 1.4 | 39.0 ± 0.7 | 10.9 ± 5.1 | 29.8 ± 2.1 |
| | | 400 | 40.9 ± 1.0 | 40.5 ± 0.7 | 11.8 ± 5.5 | 34.4 ± 2.6 |

Table 2: Values and security holes of strategies in a changed graph $G_2$ optimized from old strategy or from scratch. Initialization in $\sigma_1$ leads to higher DVal with smaller security holes in fewer iterations, unless the changes are too large.

malize all results by $100/\alpha_{\max}$ for each $G_2$ and its $\alpha_{\max}$.

*Variable edge length*   As in the previous case, the length of each edge is increased/decreased by CS% or kept unchanged (with the same probability).

*Removed edges*   We randomly delete CS edges so that $G_2$ remains strongly connected.

For each CS, we generate 10 modified graphs $G_2$. For every $G_2$, we take the highest value and security gap from 10 optimization trials with 0, 50, 100, 200, and 400 optimization steps initiated in $\sigma_1$ (recall that the optimization step in our algorithm uses noising and hence the output is different for each of the 10 trials). We report the means and standard deviations over all 10 modified graphs $G_2$. The same statistics are reported for the runs that start from random initialization instead of from $\sigma_1$. This is repeated for every CS. Hence, for each line of Tab. 2, we run $2 \times 100$ optimization trials.

For setup details, see Brázdil et al. [2022].

**Summary**   All experiments unanimously confirm that, for small CS, the initialization in $\sigma_1$ leads to higher DVal and much smaller security holes in fewer iterations. For small CS, strategies obtained after 50 optimization steps from $\sigma_1$ are not outperformed even by 400 steps of optimization initiated in a randomly chosen strategy. Only for large CS in edge length, the optimizations initiated in a random strategy reach higher values than the optimization initiated in $\sigma_1$. However, the security gap is huge.

In all our experiments, the average time needed for performing one optimization step is 110 milliseconds, which is sufficient for performing our algorithm on the fly.

**Mitigating Security Holes**   The conditions enabling randomized strategy switching are satisfied for all $\sigma_2$ summarized in utility changes of Tab. 2. We have that $\mathrm{DVal}_{G_1}(\sigma_1) = 42.1$ and $\mathrm{DVal}_{G_2}(\sigma_1) = 40.9$, which means $\mathrm{AVal}_{G_1}(\sigma_1) = 57.9$ and $\mathrm{AVal}_{G_2}(\sigma_1) = 59.1$. By Theorem 4, the security hole can be reduced arbitrarily close to 1.2 for *all* $\sigma_2$, including those constructed from randomly chosen initial strategies. Note that the improvement is *significant* in almost all cases.

# 6   CONCLUSIONS

Our experiments show that our strategy adaptation algorithm is sufficiently efficient to be run on-the-fly, outperforming the best existing strategy synthesis algorithm REGSTAR by three orders of magnitude. Furthermore, the experiments demonstrate the effectiveness of the designed methods for preventing and mitigating security holes.

An interesting open question is whether the Defender can effectively decrease the potential negative impact of environmental changes by *preventive* adaptations of its current strategy. This approach is applicable in cases when the probability of these changes happening in a near future is known.

### Acknowledgements

Research was sponsored by the Army Research Office and was accomplished under Grant Number W911NF-21-1-0189 and from Operational Programme Research, Development and Education – Project Postdoc2MUNI No. CZ.02.2.69/0.0/0.0/18_053/0016952.

*Disclaimer.*   The views and conclusions contained in this document are those of the authors and should not be interpreted as representing the official policies, either expressed or implied, of the Army Research Office or the U.S. Government. The U.S. Government is authorized to reproduce and distribute reprints for Government purposes notwithstanding any copyright notation herein.

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
