# OpenReview forum: "On-the-fly Adaptation of Patrolling Strategies in Changing Environments"
_auai.org/UAI/2022/Conference — UAI 2022 Oral_

### Official Review · Reviewer_PU5Z · 2022-04-12

**Q2(1) Originality/Novelty:** 3
**Q2(2) Significance/Impact:** 3
**Q2(3) Correctness/Technical Quality:** 3
**Q2(6) Clarity Of Writing:** 3
**Q6 Overall Score:** 7
**Q8 Confidence In Your Score:** 3

**Q1 Summary And Contributions:**

The paper aims to develop methods for computing strategies for adversarial patrolling games in an environment where traversal times of the edges of the graph and the target costs change. The paper introduces a formal definition for changing environments in patrolling games and develops an algorithm (by modifying a method for static environments). The paper also discusses how to avoid/mitigate “security holes” which are new attack possibilities occurring due to change of strategies.

**Q2 Assessment Of The Paper:**

More detailed information regarding each of these aspects is given below:

**Q2(4) Quality Of Experiments (Optional):**

3: Good: The experimental evaluation is adequate, and the results convincingly support the main claims.

**Q2(5) Reproducibility:**

1: Poor: Key details (e.g., proof sketches, experimental setup) are incomplete/unclear, or key resources (e.g., proofs, code, data) are unavailable.

**Q3 Main Strengths:**

The paper provides formal definition for a problem that was not studied previously and a method to solve this problem. It points out what kind of problems that may rise while changing strategies (i.e. security holes) and provides solutions to them. The experiments show that the introduced method performs better than the currently best existing method.

**Q4 Main Weakness:**

The paper is not very hard to follow in general but from time to time it is hard to keep track of all the symbols and definitions. In the beginning of the paper, the main idea of the patrolling games was also not clear, since the paper starts directly with many definitions instead of some intuitive explanation about these games. For someone not very familiar with the patrolling games, it may be hard to understand what is happening until the motivation for dynamic version of the problem is introduced with some examples.

Even though the formal definitions seem to be comprehensive, the developed algorithm is very briefly explained in the paper, and it seems to be a modified version of an existing method.


**Q5 Detailed Comments To The Authors:**

It could be nice to have a few sentences about the motivation about patrolling games in general or an example application before moving on to the definitions including symbols in the beginning.

The sentence defining protection value was a bit confusing, adding commas may make it more clear.


**Q7 Justification For Your Score:**

The paper defines a new problem formally and introduces an algorithm for it. It also points out what problems may rise during strategy change and provides some methods to prevent/mitigate. The algorithm is based on an existing method but seems to modify it from different aspects. The experimental results show that the new method is better. The paper is not very hard to follow but some parts of the presentation may be improved.

**Q9 Complying With Reviewing Instructions:**

1: Yes.

---

### Official Review · Reviewer_4Xk2 · 2022-04-12

**Q2(1) Originality/Novelty:** 3
**Q2(2) Significance/Impact:** 2
**Q2(3) Correctness/Technical Quality:** 3
**Q2(6) Clarity Of Writing:** 3
**Q6 Overall Score:** 7
**Q8 Confidence In Your Score:** 3

**Q1 Summary And Contributions:**

The paper presents a technique for changing the behaviour of a defending agent in a patrolling game when the environment changes. The key issue noted is that an optimal strategy for each environment may create a gap for attacks that start before the change and end after it. The technique developed involves modifying the Regstar method to adapt the current strategy rather than computing a completely different strategy. Results show this technique to outperform restarting and to be efficient.

**Q2 Assessment Of The Paper:**

More detailed information regarding each of these aspects is given below:

**Q2(4) Quality Of Experiments (Optional):**

3: Good: The experimental evaluation is adequate, and the results convincingly support the main claims.

**Q2(5) Reproducibility:**

3: Good: Key resources (e.g., proofs, code, data) are available and key details (e.g., proofs, experimental setup) are sufficiently well-described for competent researchers to confidently reproduce the main results.

**Q3 Main Strengths:**

The paper presents an interesting new problem and analyses it and problems in the domain.

The technique developed significantly outperforms the best existing approach to the static problem adapted in the most straightforward way to the new problem.

**Q4 Main Weakness:**

While the problem is clearly of practical importance, it is perhaps rather niche academically and may be of limited interest to the full conference.

**Q5 Detailed Comments To The Authors:**

The introduction lacks any references - it would be useful to provide a source for claims such as strategies use random moves, or the complexity results. To be clear, I'm not disputing such claims, but it would be normal practice to back them up with evidence at the point of use.

The introduction nicely motivates the need for new work in the area, by mentioning real-world scenarios where the problem occurs and providing a concrete example of why a naive approach may fail to ensure security.

It would be interesting to know how much of the analysis holds for multiple graph changes, i.e. G1->G2->G3->... particularly in the case that are more frequent than an attack time, i.e. the attack may span multiple changes, not just a single one. Similarly, would these results still hold if the change from G1->G2 was related the change from G2->G3? (e.g. a constant 'direction' of change over multiple changes or a repeated oscillation between a pair of graphs.)

**Q7 Justification For Your Score:**

The paper presents an interesting new problem and solution to it. The analysis and evaluation is good, and shows the need for the technique rather than simply adopting the static method.

**Q9 Complying With Reviewing Instructions:**

1: Yes.

---

### Official Review · Reviewer_xW1W · 2022-04-13

**Q2(1) Originality/Novelty:** 2
**Q2(2) Significance/Impact:** 2
**Q2(3) Correctness/Technical Quality:** 3
**Q2(6) Clarity Of Writing:** 3
**Q6 Overall Score:** 6
**Q8 Confidence In Your Score:** 3

**Q1 Summary And Contributions:**

The work proposes a method for patrolling changing environments, that is the case when there are sudden changes in the costs and connectivity of the graph representing the environment. Basically it tries to minimize security holes that could arise in these changes.
It presents an experimental comparison with the previous work that seems to be the state of the art for this problem, showing superior performance for static environments; and an experimental evaluation in a changing environment.

**Q10 Ethical Concerns (Optional):**

Not ethical concerns.

**Q2 Assessment Of The Paper:**

More detailed information regarding each of these aspects is given below:

**Q2(4) Quality Of Experiments (Optional):**

2: Fair: The experimental evaluation is weak: important baselines are missing, or the results do not adequately support the main claims.

**Q2(5) Reproducibility:**

2: Fair: Key resources (e.g., proofs, code, data) are unavailable but key details (e.g., proof sketches, experimental setup) are sufficiently well-described for an expert to confidently reproduce the main results.

**Q3 Main Strengths:**

It seems one of the first approaches for solving the patrolling problem in changing environments.

The experimental results in an static environment are superior to REGSTAR (UAI 2021).

**Q4 Main Weakness:**

No comparison in the case of changing environemnts with previous work, or al least REGSTAR (as a kind of baseline).

**Q5 Detailed Comments To The Authors:**

Suggest to include the algorithm in the main paper.

Compare with previous works / REGSTAR for the chaning environment.

**Q7 Justification For Your Score:**

It solves the case of changing environments in the patrolling domnain, a problem not solved in general before and with potential practical applications. It lacks a comparison to previous work for the changing environment experiment.

**Q9 Complying With Reviewing Instructions:**

1: Yes.

---

### Official Review · Reviewer_sgYP · 2022-04-14

**Q2(1) Originality/Novelty:** 3
**Q2(2) Significance/Impact:** 2
**Q2(3) Correctness/Technical Quality:** 3
**Q2(6) Clarity Of Writing:** 3
**Q6 Overall Score:** 5
**Q8 Confidence In Your Score:** 3

**Q1 Summary And Contributions:**

This paper attempted to design adaptive patrolling strategies when the environments such as the importance of guarded targets are changing. The authors formulate this issue as a strategy switching within dynamic patrolling graphs. Finally, synthesis experiments demonstrate its effectiveness.

**Q2 Assessment Of The Paper:**

More detailed information regarding each of these aspects is given below:

**Q2(4) Quality Of Experiments (Optional):**

2: Fair: The experimental evaluation is weak: important baselines are missing, or the results do not adequately support the main claims.

**Q2(5) Reproducibility:**

3: Good: Key resources (e.g., proofs, code, data) are available and key details (e.g., proofs, experimental setup) are sufficiently well-described for competent researchers to confidently reproduce the main results.

**Q3 Main Strengths:**

This paper attempted to design adaptive patrolling strategies when the environments such as the importance of guarded targets are changing. The authors formulate this issue as a strategy switching within dynamic patrolling graphs. Finally, synthesis experiments demonstrate its effectiveness.

Main Strengths:

1. This paper is organized well and easy to follow. The authors provide some examples and plots to illustrate their ideas, which contributes to the reading and understanding of this work.
2. The motivation is interesting and splashy.


**Q4 Main Weakness:**

The experiments seem to be weak due to the lacks of comparative methods and sufficient ablation ways. Can the fast transform algorithm between graphs be used as a comparative approach?



**Q5 Detailed Comments To The Authors:**

Doubts and Suggestions:

1. I am curious about the (running) time it takes to adjust the strategy after each environment change.

2. It is better to add a specific algorithmic procedure for the proposed methods.

**Q7 Justification For Your Score:**

Overall, I cannot judge the reproducibility and practical value of this paper, due to the lack of clear algorithm flow and weak experimental scheme. However, I have to admit the splashy motivation and specific formulation of the problem and solutions. Therefore, I take a borderline score towards acceptance.

**Q9 Complying With Reviewing Instructions:**

1: Yes.

---

### Official Review · Reviewer_kfvo · 2022-04-14

**Q2(1) Originality/Novelty:** 3
**Q2(2) Significance/Impact:** 3
**Q2(3) Correctness/Technical Quality:** 3
**Q2(6) Clarity Of Writing:** 3
**Q6 Overall Score:** 7
**Q8 Confidence In Your Score:** 4

**Q1 Summary And Contributions:**

The paper presents an approach to dynamically finding patrolling strategies in changing (graph) environments.
Specifically, the paper presents a characterization of such environments, studies the security holes that are generated by the modifications to the settings, and introduces a variant of a known algorithm for efficiently calculating patrolling strategies in the new environment (taking into account the patrolling strategies in the old environment).

**Q2 Assessment Of The Paper:**

More detailed information regarding each of these aspects is given below:

**Q2(4) Quality Of Experiments (Optional):**

3: Good: The experimental evaluation is adequate, and the results convincingly support the main claims.

**Q2(5) Reproducibility:**

4: Excellent: Key resources (e.g., proofs, code, data) are available and key details (e.g., proof sketches, experimental setup) are comprehensively described for competent researchers to confidently and easily reproduce the main results.

**Q3 Main Strengths:**

The paper presents a nice idea (that of changing environments in patrolling) that could foster further investigations in the specific subfield of patrolling games.
The paper is clearly written and makes nice use of examples.

**Q4 Main Weakness:**

I am not convinced of the frequent references to supplementary material. After reading the paper, my feeling is that the paper is self-contained as long as the general ideas are concerned, but that most of the details cannot be understood without heavily resorting to the supplementary material. The current paper seems like an executive summary of a longer document that ideally includes both paper and supplementary material.
Although I am personally not in favor of this approach, I understand that it is allowed by the conference's rules.

**Q5 Detailed Comments To The Authors:**

Authors assume that the frequency with which the environment changes is low enough to allow the current patrolling strategy to reach a "stable" state. This should be stated explicitly. If the frequency of changes is higher, it would be possible to have attacks that exploit multiple transitions, from G_{1} to G_{2} to G_{3}...

Why do authors consider only topological and utility changes? Motivation should be provided. Changes in the sets of vertices and targets (e.g., the addition of a new target) could be also plausible in applications.

From the data reported in Table 2, it seems that large changes in utility provide better values for the Defender than small changes. At a first sight, this appears counterintuitive. Do the authors have any explanation?

One open question is whether the Defender can take any advantage of knowing that the environment could change (without knowing *how* it would change, as the authors suggest in Section 6) in order to develop better strategies \sigma_{1}. For instance, if the Defender knows that G_{1} could change and there is a tie between some patrolling strategies (i.e., they all provide the same expected value), it could choose the one with fewer dead-ends in order to better react to possible changes in the environment.

Page 2: in the first t - t_{1} -> in the last t - t_{1} (?)

**Q7 Justification For Your Score:**

The major contribution of the paper is conceptual and relative to the characterization of changing patrolling environments. Algorithmic contributions are more incremental but significant.
The paper appears sound and is well written.
Experiments are adequate and code is provided.

**Q9 Complying With Reviewing Instructions:**

1: Yes.

---

### Decision · Program_Chairs · 2022-05-15

**Decision:**

Accept (Oral)

**Comment:**

Meta Review: The reviewers reach a consensus on the acceptance. The authors are encouraged to take all the comments into consideration and further improve the paper in the camera ready.